# *Agrobacterium tumefaciens*-Mediated Gene Transfer in a Major Human Skin Commensal Fungus, *Malassezia globosa*

Otomi Cho [1], Yasuhiko Matsumoto [1], Tsuyoshi Yamada [2] and Takashi Sugita [1,*]

[1] Department of Microbiology, Meiji Pharmaceutical University, 2-522-1 Noshio, Kiyose, Tokyo 204-8588, Japan
[2] Teikyo University Institute of Medical Mycology, 359 Otsuka, Hachioji, Tokyo 192-0395, Japan
* Correspondence: sugita@my-pharm.ac.jp

**Abstract:** Although the fungal microbiome in human skin mainly comprises lipophilic yeasts, including *Malassezia* species, these microorganisms can cause various dermatitis conditions, including pityriasis versicolor, seborrheic dermatitis, folliculitis, and atopic dermatitis, depending on the host condition. Both *Malassezia globosa* and *Malassezia restricta* are major species implicated in *Malassezia*-related dermatitis. However, the pathogenicity of these microorganisms has not been revealed at the genetic level owing to the lack of a genetic recombination system. Therefore, we developed a gene recombination system for *M. globosa* using *Agrobacterium tumefaciens*-mediated gene transfer of the target gene *FKB1*, which encodes the FKBP12 protein that binds the calcineurin inhibitor tacrolimus. The wild-type strain of *M. globosa* was sensitive to tacrolimus, whereas the *FKB1* deletion mutant was resistant to tacrolimus. Reintroduction of *FKB1* into the *FKB1* deletion mutant restored wild-type levels of susceptibility to tacrolimus. Moreover, an *FKB1-eGFP* fusion gene was generated and expression of this fusion protein was observed in the cytoplasm. This newly developed gene recombination system for *M. globosa* will help further our understanding of the pathogenesis of *M. globosa*-related dermatitis at the genetic level.

**Keywords:** *Agrobacterium tumefaciens*-mediated gene transfer; *FKB1*; gene recombination; *Malassezia globosa*; skin microbiome




## 1. Introduction

The human skin microbiome comprises bacteria, fungi, and viruses. In the bacterial microbiome, the genera *Cutibacterium* (formerly *Propionibacterium*), *Corynebacterium*, and *Staphylococcus* are predominant, whereas the skin fungal microbiota primarily comprises *Malassezia* species regardless of the skin area. *Malassezia* species commonly inhabit sebum-rich areas, such as the scalp, face, and neck, because they use sebum as a nutrient source [1,2]. The genus *Malassezia* includes 18 species, of which nine species (*Malassezia dermatis*, *Malassezia furfur*, *Malassezia globosa*, *Malassezia japonica*, *Malassezia obtusa*, *Malassezia restricta*, *Malassezia slooffiae*, *Malassezia sympodialis*, and *Malassezia yamatoensis*) inhabit the human skin. The remaining *Malassezia* species are commonly found on the skin of other animals. The fungal microbiome in humans primarily comprises *M. globosa* and *M. restricta*. Although *Malassezia* species are commensal microorganisms present on human skin, they also cause seborrheic dermatitis, including dandruff, pityriasis versicolor, and *Malassezia* folliculitis, and can exacerbate atopic dermatitis [3–5]. Although they inhabit human skin, the abundance of *M. restricta* and *M. globosa* differs depending on the skin disease. *M. restricta* is more abundant than *M. globosa* in the lesions of patients with seborrheic and atopic dermatitis, whereas *M. globosa* is more frequently detected in patients with pityriasis versicolor. *Malassezia* species produce many secretory lipases (thirteen in *M. globosa* and nine in *M. restricta*) that hydrolyze diglycerides or triglycerides in the sebum into glycerin and free fatty acids [6]. These metabolites are used as nutrients by many skin microbiota, including *Malassezia*. Oleic acid present in free fatty acids can induce seborrheic dermatitis;

thus, lipases have been considered as virulence factors of *Malassezia* [7]. Anti-*Malassezia* manganese superoxide dismutase and cyclophilin-specific immunoglobulin E antibodies are produced in the sera of patients with atopic dermatitis, suggesting that the presence of *Malassezia* exacerbates this disease [8,9]. Pityriasis versicolor is characterized by scaly hypo- or hyperpigmented lesions usually affecting the trunk. Notably, a large amount of hyphae are observed in patient lesions, suggesting that dimorphic conversion may be responsible for the development of pityriasis versicolor [10]. Various factors have been suggested to be involved in *Malassezia*-related skin dermatitis. Deletion of the genes involved in virulence may help elucidate the functions of the relevant genes.

To date, gene deletion in *Malassezia* has only been conducted in *M. furfur*, *M. sympodialis*, and *M. pachydermatis*; gene deletion methods using clinically important species, *M. globosa* and *M. restricta*, have not yet been established [11–13]. However, we recently established a gene recombination system in *M. restricta* using *Agrobacterium tumefaciens*-mediated gene transfer (ATMT) [14]. Although *Agrobacterium tumefaciens*-mediated gene transfer (ATMT) was developed as a method for gene introduction into plant cells, it is possible to introduce genes into a wide range of hosts, including fungi. When *Agrobacterium tumefaciens* infects a fungus, it transfers exogenous genes into the host cell by using the property of integrating transferred DNA (T-DNA), part of the plasmid, into the chromosomal DNA of the host. The ATMT method has the advantage of simplicity compared to direct gene transfer methods such as the electroporation and particle gun methods [15]. To elucidate the pathogenicity or virulence of the causative agents of *Malassezia*-associated skin diseases, it will be important to establish a gene recombination system for *M. globosa*; targeting the genes of known function will be useful in establishing such a system for *M. globosa*.

The *FKB1* gene, which encodes the 12-kDa FK506-binding protein (FKBP12), a protein that binds the calcineurin inhibitor tacrolimus, was previously deleted in *M. sympodialis* and *M. restricta* [12,14]. Since *Malassezia* species are sensitive to tacrolimus, deletion of the *FKB1* gene results in resistance to tacrolimus. Therefore, it is easy to evaluate the function of the deleted gene. In the present study, we aimed to delete the *FKB1* gene in *M. globosa* using ATMT and constructed an *FKB1*-enhanced green fluorescent protein (*eGFP*) fusion gene. The developed gene recombination system in this study will help further our understanding of the pathogenesis of *M. globosa*-related dermatitis at the genetic level.

## 2. Materials and Methods

### 2.1. Strains and Media

*M. globosa* CBS 7966 (type strain of the species) was obtained from the CBS-KNAW culture collection (https://wi.knaw.nl/ accessed on 1 July 2022) and maintained on modified Leeming and Notman agar (mLNA; 10 g/L polypeptone, 10 g/L glucose, 2 g/L yeast extract, 8 g/L ox bile, 10 mL/L glycerol, 0.5 g/L glycerol monostearate, 5 mL/L Tween-60, 20 mL/L olive oil, and 15 g/L agar) at 32 °C. The strains generated in this study are listed in Table 1.

**Table 1.** *Malassezia globosa* strains used in this study.

| *M. globosa* Strains | Relevant Genotype | Background | Reference |
|---|---|---|---|
| CBS 7966 | Wild-type | | |
| Mg::NAT-1 | *NAT1* | CBS 7966 | This study |
| Mg::NAT-2 | *NAT1* | CBS 7966 | This study |
| Mg::NAT-3 | *NAT1* | CBS 7966 | This study |
| Mg Δ*fkb1*::NAT-1 | Δ*fkb1::NAT1* | CBS 7966 | This study |
| Mg Δ*fkb1*::NAT-2 | Δ*fkb1::NAT1* | CBS 7966 | This study |
| Mg Δ*fkb1*::NAT-3 | Δ*fkb1::NAT1* | CBS 7966 | This study |
| Δ*fkb1* + FKB1 | Δ*fkb1::NAT1 FKB1::hph* | CBS 7966 Δ*fkb1* | This study |
| Δ*fkb1* + FKB1-eGFP1 | Δ*fkb1::NAT1 FKB1::eGFP::hph* | CBS 7966 Δ*fkb1* | This study |
| Δ*fkb1* + FKB1-eGFP2 | Δ*fkb1::NAT1 FKB1::eGFP::hph* | CBS 7966 Δ*fkb1* | This study |

### 2.2. Construction of M. globosa Expressing the NAT1 Gene Using ATMT

ATMT was performed using a protocol previously developed by Cho et al. [14], Matsumoto et al. [16], and Yamada et al. [17], with minor modification. The binary vector pAg1-N-terminal acetyl transferase (*gNAT1*) was constructed using the primers FK5 and MgPactin1–3 (Table S1) (Figure 1A). The promoter region of the actin gene of *M. globosa* (MgPactin) was incorporated upstream of the *NAT1* gene. pAg1-*gNAT1* was introduced via electroporation into *A. tumefaciens* EHA105, which was then cultured in 2 × YT agar (16 g/L tryptone, 10 g/L yeast extract, 5 g/L NaCl, and 15 g/L agar) containing 50 µg/mL kanamycin (Fujifilm, Osaka, Japan) at 27 °C for 2 days. The cell concentration was adjusted to OD630 = 1 in *Agrobacterium* induction medium [AIM; 2.05 g/L $K_2HPO_4$, 1.45 g/L $KH_2PO_4$, 0.15 g/L NaCl, 0.5 g/L $MgSO_4 \cdot 7H_2O$, 0.1 g/L $CaCl_2 \cdot 6H_2O$, 0.0025 g/L $FeSO_4 \cdot 7H_2O$, and 0.5 g/L $(NH_4)_2SO_4$ supplemented with 40 mM 2-(*N*-morpholino)ethanesulfonic acid (pH 5.3), 10 mM glucose, and 0.5% (w/v) glycerol] containing 200 µg/mL acetosyringone (Fujifilm) and incubated at 27 °C for 6 h. *M. globosa* (100 µL; >2 × $10^8$ cells/mL) and *Agrobacterium* suspensions were spread onto sterilized nylon membranes (Hybond N+ membranes; Amersham Biosciences, Buckinghamshire, United Kingdom). The membranes were placed on AIM containing 200 µg/mL acetosyringone and 1.5% agar and incubated at 25 °C for 2 days. Transformants were obtained from mLNA supplemented with 200 µg/mL cefotaxime sodium and 100 µg/mL nourseothricin (Jena Bioscience, Jena, Germany) and further re-grown on mLNA containing 100 µg/mL nourseothricin. Polymerase chain reaction (PCR) was performed using the primers NAT-F/R and Actin-F/R (Table S1), which confirmed the introduction of the *NAT1* gene into the *M. globosa* genome.

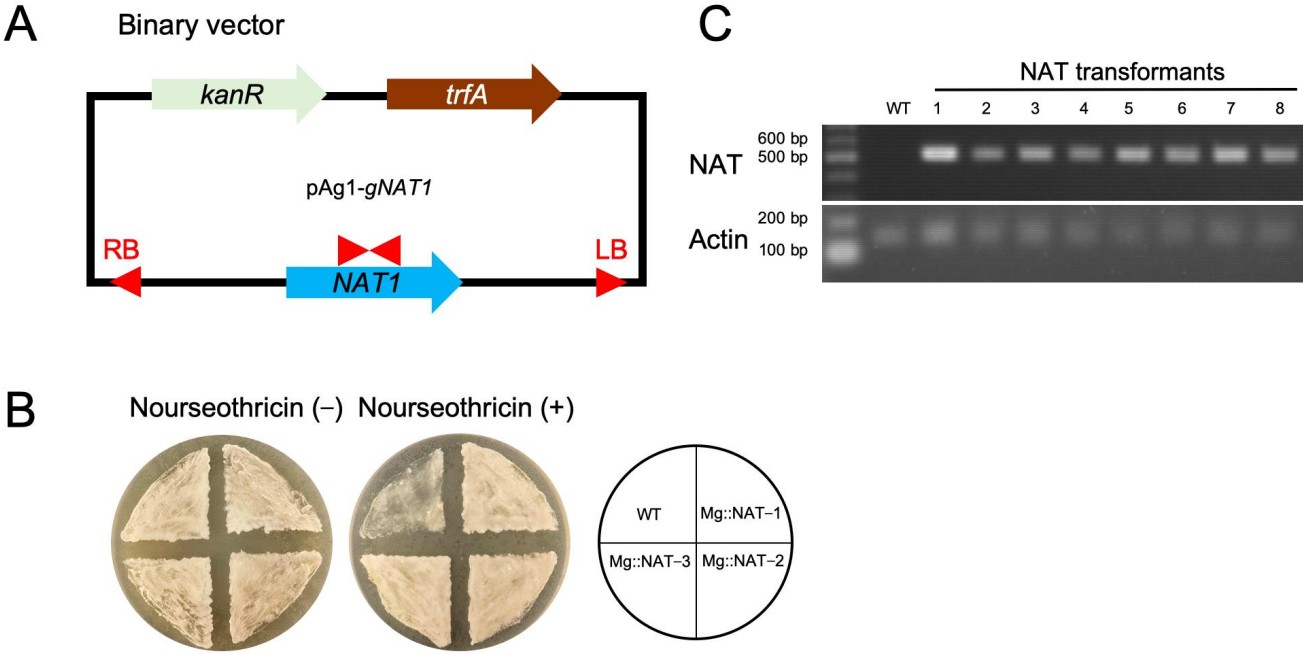

**Figure 1.** *Agrobacterium tumefaciens*-mediated transformation. (**A**) Schematic representation of the binary vector containing *NAT1* introduced into *Malassezia globosa*. The promoter sequence of the actin gene of *M. globosa* was incorporated upstream of the *NAT1* gene, and the termination sequence of the *TRP1* gene of *Cryptococcus neoformans* was incorporated downstream of the *NAT1* gene. Red arrowheads indicate primer-binding sites. (**B**) Sensitivity of the WT strain and Δ*fkb1 M. globosa* mutant on modified Leeming and Notman agar containing nourseothricin. (**C**) Polymerase chain reaction confirmation of the introduction of the *NAT1* gene into the *M. globosa* genome. *NAT1*, nourseothricin resistance gene; WT, wild-type strain; Δ*fkb1, FKB1*-knockout mutant.

### 2.3. Targeted Gene Replacement in M. globosa via ATMT

The sequence of the target gene, *FKB1*, in the *M. globosa* CBS7966 (RPJNA27973) genome is similar to that of *Candida albicans* (AAA34367). Reciprocal basic local alignment search tool protein analysis of the *C. albicans* retinol-binding protein 1 gene revealed that FKBP12 was encoded by MGL_1262. Flanking regions of approximately 1.5 kb upstream and downstream of the *FKB1* gene were introduced into the pAg1-*gNAT1* vector; the vector and two flanking regions were amplified by PCR using the primers FK1–8 (Table S1) (Figure 2A). Amplicons were cloned into competent *Escherichia coli* (Thermo Fisher Scientific, Tokyo, Japan) using the In-FusionHD™ Cloning Kit (Takara, Shiga, Japan) according to the manufacturer's instructions, and transformants were confirmed via PCR using the primers FK9 and FK10 (Table S1). The method of introduction of pAg1-Δ*fkb1*::*NAT1* into the *M. globosa* genome was similar to that of pAg1-*gNAT1* described above. Deletion of *FKB1* in the *M. globosa* genome was confirmed via PCR using the primers FK11–16, FKB1-F/R, and NAT-F/R (Table S1). *M. globosa* CBS 7966 and the *FKB1* deletion mutant (Δ*fkb1*) were grown on mLNA agar with or without 100 µg/mL tacrolimus at 32 °C for 4 days.

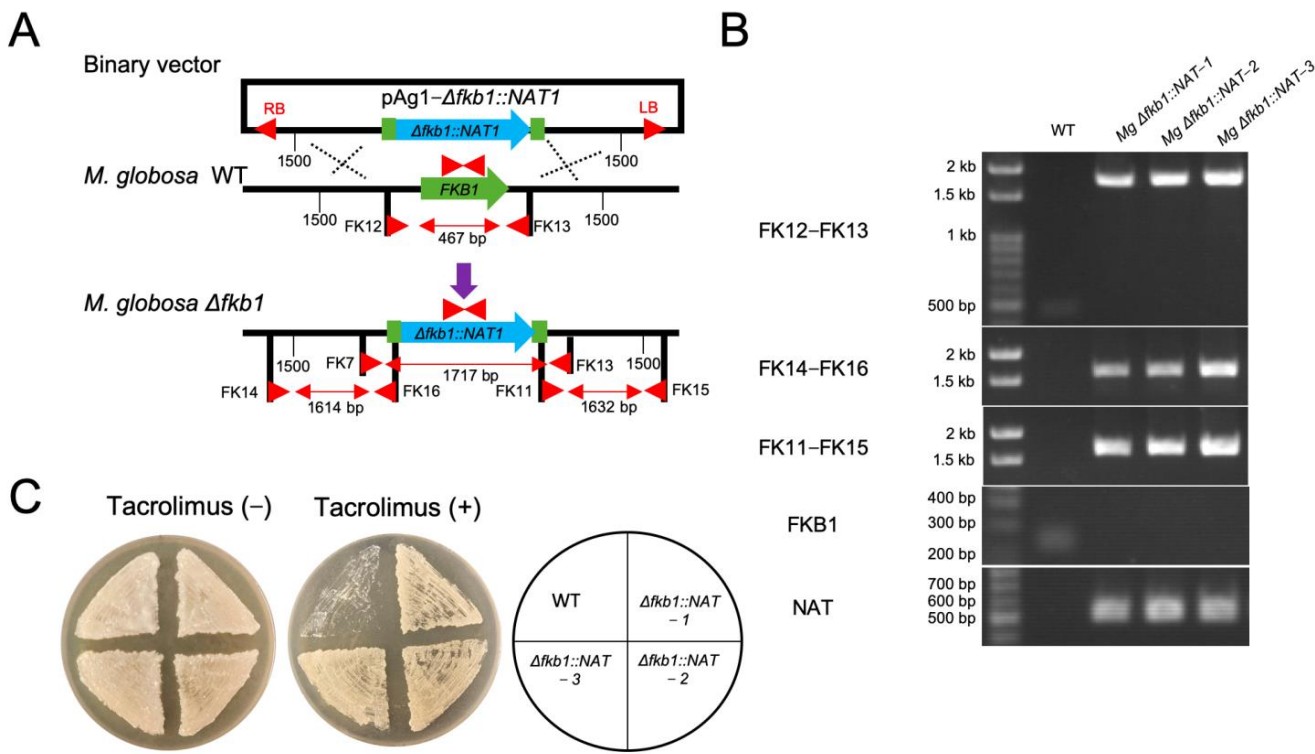

**Figure 2.** *FKB1* gene replacement in *Malassezia globosa*. *FKB1* in *M. globosa*. (**A**) Schematic representation of targeted gene replacement of the *FKB1* gene in *M. globosa*. Red arrowheads indicate primer-binding sites. (**B**) Confirmation of the *M. globosa* Δ*fkb1* strain by PCR. (**C**) Sensitivity of the WT strain and the Δ*fkb1* mutant to tacrolimus. The medium contained 100 µg/mL of tacrolimus. *NAT1*, nourseothricin resistance gene; WT, wild-type strain; Δ*fkb1*, *FKB1*-deletion mutant.

### 2.4. Reintroduction of the Target Gene into the M. globosa Δfkb1 Mutant

*FKB1* was reintroduced into the *M. globosa* Δ*fkb1* mutant as previously described [14]. Briefly, flanking regions of approximately 200 bp downstream of *FKB1* and hygromycin B phosphotransferase gene (*hph*) were introduced into the vector pAg1-Δ*fkb1*::*NAT1* (Figure 3A, and the binary vector pAg1-Δ*fkb1* + *FKB1* was transformed into *A. tumefaciens*. Introduction of *FKB1* into the genome of the transformants was confirmed via PCR using the primers FK14–16, FKB1-F/R, NAT-F/R, and Hyg-F/R (Table S1).

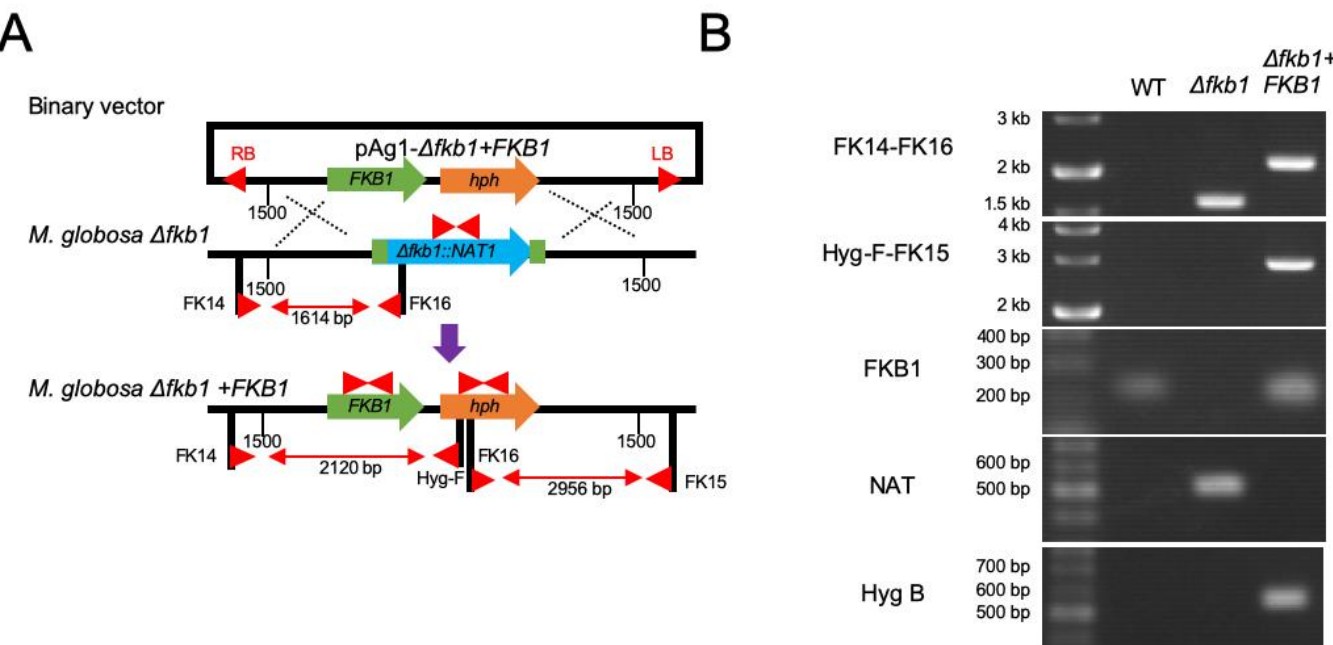

**Figure 3.** Reintroduction of the *FKB1* gene into the *Malassezia globosa* Δ*fkb1* mutant. (**A**) Schematic representation of reintroduction of *FKB1* into the *M. globosa* Δ*fkb1* mutant. Red arrowheads indicate primer-binding sites. (**B**) Polymerase chain reaction confirmation of reintroduction of the *FKB1* gene into the *M. globosa* Δ*fkb1* mutant. *NAT1*, nourseothricin resistance gene; WT, wild-type strain; Δ*fkb1*, *FKB1*-knockout mutant; *hph,* hygromycin B phosphotransferase.

### 2.5. Construction of the FKB1-eGFP Fusion Gene in M. globosa

The binary vectors pAg1-Δ*fkb1* + *FKB1-eGFP1* and pAg1-Δ*fkb1* + *FKB1-eGFP2* were generated to reintroduce the target gene into the Δ*fkb1* mutant (Figure 4A), and the binary vector was transformed into *A. tumefaciens* as described above. The introduction of the *FKB1-eGFP* fusion genes into the genome of the transformants was confirmed via PCR using the primers FK14–16, FKB1-F/R, NAT-F/R, Hyg-F/R, and eGFP-F/R (Table S1). The selected transformants were grown on mLNA plates containing 100 μg/mL nourseothricin or 100 μg/mL hygromycin B (Nacalai Tesque, Kyoto, Japan). eGFP expression in Δ*fkb1*, Δ*fkb1* + *FKB1*, and Δ*fkb1* + *FKB1-eGFP M. globosa* mutants was observed using a fluorescence microscope (BX61; Olympus Corporation, Tokyo, Japan).

### 2.6. Calcineurin Inhibitor Susceptibility

The minimum inhibitory concentrations (MICs) of the calcineurin inhibitors tacrolimus and cyclosporine A were determined using the broth microdilution method in accordance with CLSI M27-A3 as described by Rojas et al. [18]. *M. globosa* cells were incubated for 6 days at 32 °C in the presence of 0.03–100 μg/mL of tacrolimus (Fujifilm) or cyclosporine A (Fujifilm) in 96-well microtiter plates. The MIC was defined as the lowest concentration that completely inhibited growth.

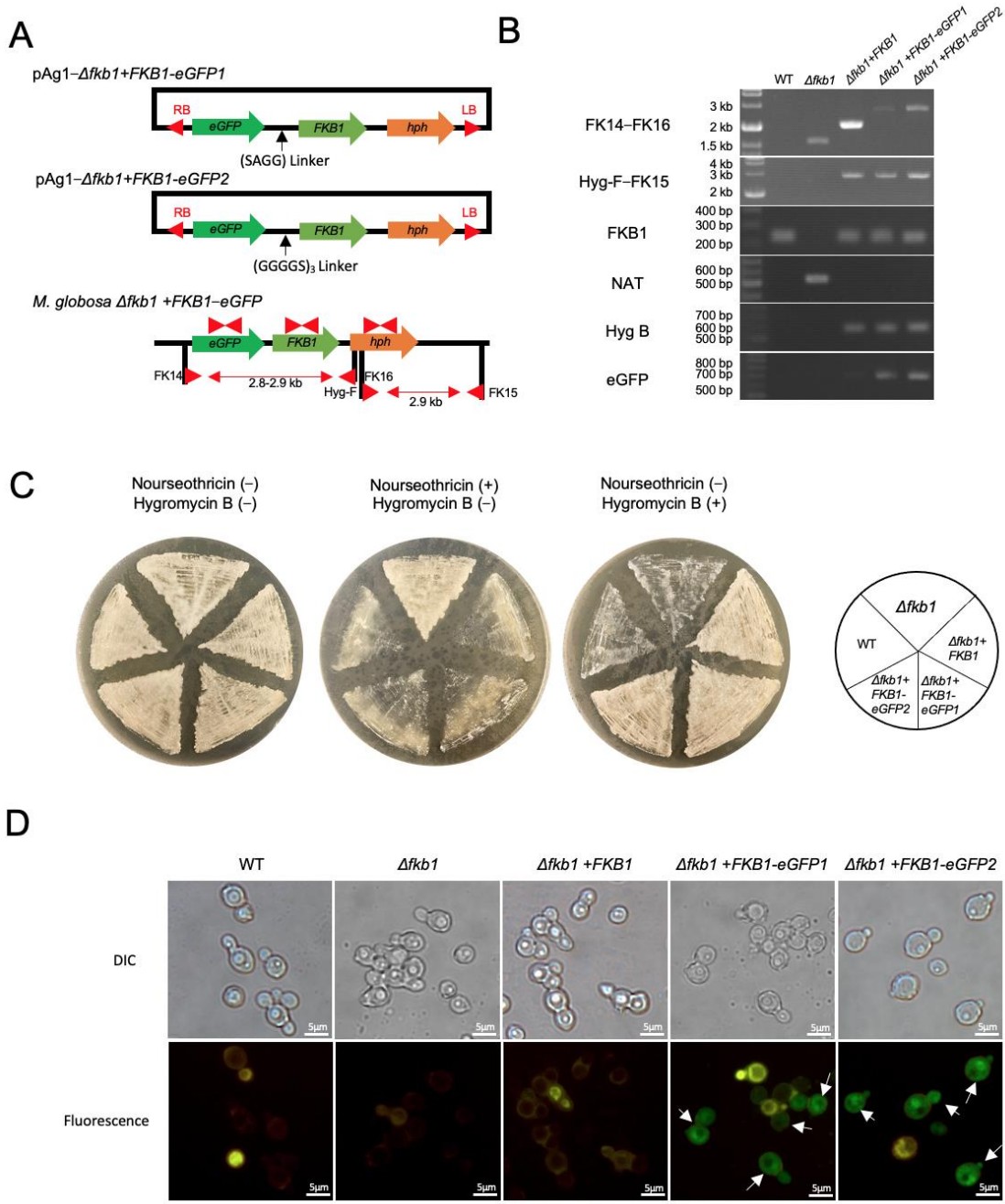

**Figure 4.** Construction of the *FKB1-eGFP* fusion gene and fluorescence microscopy. (**A**) Schematic representation of the *FKB1-eGFP* fusion gene. Two types of linker, SAGG or (GGGGS)₃, were used for construction. Red arrowheads indicate primer-binding sites. (**B**) PCR confirmation of reintroduction of the *FKB1-eGFP* fusion gene into the *Malassezia globosa* Δ*fkb1* mutant. (**C**) Sensitivity of WT strain and Δ*fkb1*, Δ*fkb1* + *FKB1*, and Δ*fkb1* + *FKB1-eGFP M. globosa* mutants to modified Leeming and Notman agar containing nourseothricin or hygromycin B. (**D**) Fluorescence microscopy of WT *M. globosa* and mutants (Δ*fkb1*, Δ*fkb1* + *FKB1*, and Δ*fkb1* + *FKB1-eGFP*). Images were compiled using Adobe Photoshop. Arrows indicate eGFP expression in cells (green fluorescence). Yellow fluorescence represents autofluorescence of *M. globosa* strains. Scale bars, 5 µm. *NAT1*, nourseothricin resistance gene; *eGFP*, enhanced green fluorescence protein; WT, wild-type strain; Δ*fkb1*, *FKB1*-knockout mutant.

## 3. Results

### 3.1. M. globosa Transformation Using ATMT

The binary vector pAg1-*gNAT1* used in the ATMT system is presented in Figure 1A. Transformants were obtained from mLNA containing nourseothricin (Figure 1B), and *NAT1* expression in the transformants was confirmed via PCR using *NAT1*-specific primers. Of the randomly selected eight colonies, all were positive; the wild-type strain was negative (Figure 1C).

### 3.2. FKB1 Gene Replacement in M. globosa

The binary plasmid pAg1-Δ*fkb1*::*NAT1*, which contained regions of approximately 1.5 kb upstream and downstream of the *FKB1* gene, was constructed for the *FKB1* gene deletion (Figure 2A). Of the randomly selected 24 colonies, *FKB1* was deleted in 23 transformants (95.8%) based on PCR confirmation (Figure 2B). The wild-type strain did not grow on tacrolimus-containing mLNA, whereas the Δ*fkb1* mutant showed growth in this medium (Figure 2C).

### 3.3. Reintroduction of the FKB1 Gene into the M. globosa Δfkb1 Mutant

The *FKB1* gene was reintroduced into the *M. globosa* Δ*fkb1* mutant via ATMT using the pAg1-Δ*fkb1*+*FKB1* plasmid (Figure 3A). Of the 20 randomly selected colonies recovered from mLNA supplemented with 100 μg/mL hygromycin B, 11 transformants (55%) harbored the *FKB1* gene in the genome based on PCR confirmation (Figure 3B).

### 3.4. Construction of the FKB1-eGFP Fusion Gene and Fluorescence Microscopy

To assess the localization of FKBP12, the *M. globosa* Δ*fkb1* + *FKB1-eGFP* mutant was generated using two binary vectors (pAg1-Δ*fkb1*+*FKB1-eGFP1* and *pAg1-Δfkb1+FKB1-eGFP2*) (Figure 4A). Reintroduction of *FKB1-eGFP1* and *FKB1-eGFP2* into the transformants was confirmed via PCR (Figure 4B). The *M. globosa* Δ*fkb1* + *FKB1* and Δ*fkb1* + *FKB1-eGFP* mutants were sensitive to nourseothricin and resistant to hygromycin B (Figure 4C). eGFP expression was observed in only Δ*fkb1* + *FKB1-eGFP* mutants; yellow autofluorescence was observed in each strain (Figure 4D). Fluorescence microscopy revealed that the FKB1-eGPF fusion protein was expressed in the cytoplasm.

### 3.5. Drug Susceptibility to Calcineurin Inhibitors

The MICs of tacrolimus ranged from 0.06 to 0.12 μg/mL for the wild-type strain of *M. globosa* and were >100 μg/mL for the Δ*fkb1* mutant. The gene-complemented mutant, Δ*fkb1* + *FKB1*, had the same MIC as that of the wild-type strain. The susceptibility of the Δ*fkb1* + *FKB1*-eGFP mutant to tacrolimus was similar to that of the wild type and Δ*fkb1* + *FKB1* mutant (Table 2). However, the wild type and all mutants (Δ*fkb1*, Δ*fkb1* + *FKB1*, and Δ*fkb1* + *FKB1-eGFP*) showed no difference in susceptibility to cyclosporine A, which does not bind FKBP12 (Table 2).

**Table 2.** Drug sensitivity of *Malassezia globosa* strains to tacrolimus and cyclosporine A.

| *M. globosa* Strain | MIC (μg/mL) | |
| --- | --- | --- |
| | **Tacrolimus** | **Cyclosporin A** |
| Wild-type | 0.06–0.12 | 8–16 |
| Δ*fkb1* | >100 | 8–16 |
| Δ*fkb1* + *FKB1* | 0.06–0.12 | 8–16 |
| Δ*fkb1* + *FKB1-eGFP1* | 0.06–0.12 | 8–16 |
| Δ*fkb1* + *FKB1-eGFP2* | 0.06–0.12 | 8–16 |

## 4. Discussion

Genetic studies on *Malassezia* have not progressed as rapidly as those on *Candida* and *Cryptococcus*. In recent years, gene recombination has been successfully reported for

*M. furfur*, *M. sympodialis*, and *M. pachydermatis* using ATMT [11–13]. The main causative or exacerbating agents of *Malassezia*-related skin dermatitis are *M. restricta* and *M. globosa* [5,19–21]. Following our recently established gene recombination method for *M. restricta* [14], in the present study, we developed a similar method for *M. globosa* targeting the *FKB1* gene. The *FKB1* gene encodes FKBP12, a protein that binds the calcineurin inhibitor tacrolimus; therefore, the Δ*fkb1* mutant was no longer susceptible to tacrolimus [22,23].

The gene recombination method for *M. globosa* was essentially the same as that for *M. restricta*, with the following modifications. First, the promoter of the binary vector was changed from the *Cryptococcus neoformans* actin promoter to the *M. globosa* actin promoter. Although the *C. neoformans* actin promoter may function in *Malassezia* cells, a promoter derived from the same species was expected to function better. The *M. globosa* actin promoter was functional in *M. globosa* mutants in this study. Second, the concentration of *Malassezia* cells was critical for transformation. We optimized the ATMT protocol for *M. globosa* to use a cell concentration of $2 \times 10^8$ cells/mL, whereas the protocol for *M. restricta* uses $6 \times 10^8$ cells/mL. With cell concentrations of $<2 \times 10^8$ cells/mL, few transformants were detected. Finally, the concentration of acetosyringone (200 μg/mL) in the AIM and the incubation temperature of 25 °C were optimized for *M. globosa* to increase transformation efficiency (for *M. restricta*, 40 μg/mL of acetosyringone and 27 °C of incubation temperature were applied).

GFP fusion proteins are widely used to investigate protein localization within cells, and the expression of GFP fusion genes has been successfully observed in *Malassezia* species [24,25]. We used two linker sequences, SAGG and (GGGGS)$_3$, to create eGFP fusion genes [26]. We found that eGFP fusion genes could be created using either linker sequence in *M. globosa*.

Although *M. globosa* is implicated in all *Malassezia*-related dermatitis conditions, it is thought to be more involved in the development of pityriasis versicolor [19,27,28]. *M. globosa* is relatively more abundant in lesions of patients with pityriasis versicolor, and numerous hyphae formed by the microorganism is observed, suggesting that dimorphic conversion of *M. globosa* may be involved in the pathogenesis of dermatitis [5,20]. However, a genetic understanding of the dimorphism pathway in *M. globosa* is lacking. *M. globosa* is also responsible for exacerbating atopic dermatitis. MGL_1304 in *M. globosa* has been identified as a causative antigen of sweat allergies observed in patients with atopic dermatitis and cholinergic urticaria; the present study suggests that the skin microbiota may represent a potential allergen [29,30]. Further elucidation of the functions of dimorphism-related and allergen-encoding genes of *M. globosa* will be possible using the gene recombination method established in this study.

**Supplementary Materials:** The following supporting information can be downloaded at: https://www.mdpi.com/article/10.3390/applmicrobiol2040063/s1, Table S1. Primers used in this study.

**Author Contributions:** Conceptualization, O.C. and T.S.; methodology, O.C., Y.M. and T.Y.; investigation, O.C., Y.M. and T.Y.; data curation, O.C., Y.M., T.Y. and T.S.; writing—original draft, O.C.; writing—review and editing, Y.M., T.Y. and T.S.; visualization, O.C., Y.M. and T.Y.; project administration, T.S. All authors have read and agreed to the final version of the manuscript.

**Funding:** This work was partially supported by the Japan Society for the Promotion of Science KAKENHI (grant number JP20K07208 to O.C.).

**Institutional Review Board Statement:** Not applicable.

**Informed Consent Statement:** Not applicable.

**Data Availability Statement:** Essential data supporting the reported results are provided in the article.

**Conflicts of Interest:** The authors have no financial, consultant, institutional, and other relationships that might lead to bias or conflict of interest.

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
