# Peer review of "Agrobacterium tumefaciens-Mediated Gene Transfer in a Major Human Skin Commensal Fungus, Malassezia globosa"

_2673-8007, doi:10.3390/applmicrobiol2040063_

Round 1
Reviewer 1 Report
Considering the lack of knowledge about malasseziasis, the results are relevant and allow further studies on opportunistic fungi as reported here. The creation of new methodologies for observation of pathogenic effects is fundamental for this field.
Author Response
Reviewer 1
Considering the lack of knowledge about malasseziasis, the results are relevant and allow further studies on opportunistic fungi as reported here. The creation of new methodologies for observation of pathogenic effects is fundamental for this field.
Reply: Thank you for your comments.
We were able to develop a new genetic recombination technology for M. globosa, a major causative agent of Malassezia-related skin diseases. We believe that this technology will enable us to elucidate the onset mechanism of dermatitis at the molecular level.
Author Response
Reviewer 2
Major comments
Q1: Authors are advised to consider following for the introduction section.
As per reference, authors have already demonstrated the application of the Agrobacterium tumefaciens-mediated gene transfer in M. restricta. The work described in this article appears to be a second case study demonstrating an application of ATMS. While authors tried to demonstrate the importance of the genetic recombination in M. globose. Authors are advised to establish any other novel aspect of the work beyond just application of the same technique for a different species from same family. They may focus on describing potential challenges with application of ATMS to other species from Malassezia (M. globose) or the aspects of this work that are different from their recently published studies on M restricta.
Reply: Thank you for your comments.
As you point out, in this study the principle of the genetic recombination method for M. globosa is basically the same as that for M. restricta. The genus Malassezia includes 18 species, nine of them capable of causing dermatitis in humans. Gene recombination methods for the human-associated species M. furfur and M. sympodialis were developed in 2017 [12, 13]. However, the frequency of these two species in skin is very low. Although M. restricta and M. globosa are the major causative agents of dermatitis, nobody has succeeded in establishing a genetic recombination technology for them. As mentioned in the Discussion section, one of the reasons why genetic recombination technology for these two species has not been established is that they are difficult to culture. Therefore, we investigated culture methods for these two fungi and succeeded in optimizing a method. First, we succeeded in developing genetic recombination technology for M. restricta, and the paper was submitted to another journal and accepted. After this, we succeeded in developing a genetic recombination technology for M. globosa, and are now submitting it to this journal. Since M. restricta and M. globosa are the major causative agents of Malassezia-associated dermatitis, we believe that the successful development of genetic engineering technology for these two species is of great significance.
- globosa, in particular, is a major causative agent of pityriasis versicolor (PV). Since hyphae can be observed in clinical lesions of PV, dimorphic conversion is considered to be one of the virulence factors. The exacerbation antigen MGL_1304 for atopic dermatitis has also been identified. By establishing genetic recombination technology for M. globosa in this research, it will be possible to elucidate at the genetic level that M. globosa is involved in the exacerbation of PV and atopic dermatitis. You suggest mentioning this in the introduction, but it is already in the Discussion section. We trust this is acceptable.
Q2: Authors are advised to include description of the already available genetic recombination techniques for Malassezia narrowing down to Agrobacterium tumefaciens-mediated gene transfer. They can further elaborate on Agrobacterium tumefaciens-mediated gene transfer system by describing the basic concept and advantages of this approach (since authors have already demonstrated the application of this technique, they should be able to elaborate on this)
Reply: We have added the following sentences according to your suggestion.
Although the Agrobacterium tumefaciens-mediated gene transfer (ATMT) was developed as a method for gene introduction into plant cells, it is possible to introduce genes into a wide range of hosts, including fungi. When Agrobacterium tumefaciens infects a fungus, it transfers exogenous genes into the host cell by using the property of integrating Transferred DNA (T-DNA), part of the plasmid, into the chromosomal DNA of the host. The ATMT method has the advantage of simplicity compared to direct gene transfer methods such as the electroporation and particle gun methods [15].
- Idnurm A, Bailey AM, Cairns TC, Elliott CE, Foster GD, Ianiri G, Jeon J.
A silver bullet in a golden age of functional genomics: the impact of Agrobacterium-mediated transformation of fungi. Fungal Biol. Biotechnol. 2017, 4, 6.
Minor comments
Q3: Table 1 – What is the difference between Mg::NAT-1, Mg::NAT-2 and Mg::NAT-3 strains
Reply: In this study, three strains were randomly selected for the experiment. In the results, all strains showed the same characteristics.
Q4: Line 241-245: Introduction of e-GFP serves as another example demonstrating the application of ATMS in M. Globosa. Where GFP is a good model protein, expression of which can be easily detected. Is there any other reason for which eGFP incorporation in M. Globosa was used? Both the linkers SAGG and GGGGS seem to work. What is the reason behind trying these two linkers? And why it is significant?
Reply: GFP fusion proteins are widely used to investigate protein localization within cells. Linkers SAGG and GGGGS are most commonly used. Therefore, these two linkers were used to see which would be most suitable. Both linkers were found to be useful in this study (Figure 4).
